



# Data-driven distinction between convective, frontal and mixed extreme rainfall events in radar data

Emma Dybro Thomassen[1], Hjalte Jomo Danielsen Sørup[1], Marc Scheibel[2], Thomas Einfalt[3], Karsten Arnbjerg-Nielsen[1]

[1]Department of Environmental Engineering, Technical University of Denmark, Lyngby, 2800, Denmark
[2]Wupperverband, Wuppertal, 42289, Germany
[3]hydro & meteo GmbH, Lübeck, 23552, Germany

*Correspondence to*: Karsten Arnbjerg-Nielsen (karn@env.dtu.dk)

**Abstract.** This study examines characteristics of extreme events based on a high-resolution precipitation dataset (5-minute

temporal resolution, 1x1 km spatial resolution) over an area of 1824 km$^2$ covering the catchment of the river Wupper, North Rhine-Westphalia, Germany. Extreme events were sampled by a Peak Over Threshold method using several sampling strategies, all based on selecting an average of three events per year. A simple identification- and tracking algorithm for rain cells based on intensity threshold and fitting of ellipsoids, is developed for the study. Extremes were selected based on maximum intensities for 15-minute, hourly and daily durations and described by a set of 17 variables. The spatio-temporal

properties of the extreme events are explored by means of a principal component analysis (PCA) and a cluster analysis for these 17 variables. We found that these analyses enabled us to distinguish and characterise types of extreme events useful for urban hydrology applications. The PCA indicated between 5 and 9 dimensions in the extreme event characteristic data. The cluster analyses identified four rainfall types: convective extremes, frontal extremes, mixed very extreme events and other extreme events, the last group consisting of events that are less extreme than the other events. The result is useful for selecting

events of particular interest when assessing performance of e.g. urban drainage systems.

## 1 Introduction

Urban hydrological models of high quality are a required tool to make cities more resilient to pluvial flooding and pollution management. A key input parameter when modelling urban drainage systems is rainfall (Berndtsson and Niemczynowicz, 1988; Schilling, 1991; Thorndahl et al., 2008; Vaes et al., 2001). A common way is to use a model including a rainfall-runoff

component that uses rainfall input as either a long-term rainfall series or a design storm (Butler and Davies, 2011; Willems et al., 2012). For some applications rainfall data must be of high spatial and temporal resolution (Berndtsson and Niemczynowicz, 1988; Einfalt et al., 2004; Ochoa-Rodriguez et al., 2015; Schilling, 1991). Schilling (1991) and Einfalt et al. (2004) have proposed resolution requirements of 1-5 minute temporal resolution and 1x1 kilometre spatial resolution. For planning and design purposes these reviews suggest that at least 10-20 years of data should be available.




Inference on properties of rainfall can be based upon two types of data: rain gauge and radar data. Both types of data have significant strengths and weaknesses. Rain gauge data require less data treatment compared to radar data, and measurements are often available for longer time periods. Rain gauges measure rainfall at ground level, which is the rainfall of interest in hydrological modelling (Thorndahl et al., 2016), and often have a temporal resolution of around 1 minute (Einfalt et al., 2004).

A major weakness about rain gauge data is the lack of information on rainfall movement (mainly for convective events), spatial variation and coverage. Radar data, on the other hand, gives information about rainfall movement and spatial coverage (Thorndahl et al., 2016), and have significantly improved our understanding of how precipitation is formed (Collier, 1989). The weaknesses of radar data is that it is an indirect remote sensing measurement. Here rainfall intensities are inferred based on reflectivity with often very high uncertainties for high rainfall intensities. Furthermore, radar data is based on an

instantaneous scan of volume high above ground that is then used to represent the average rainfall intensity during the entire sampling time. This can lead to aggregation errors and might not reflect the rainfall at ground level (Einfalt et al., 2004).

Radar products have more recently become available in spatial and temporal resolution fulfilling the resolution requirement in urban hydrology, and has within the last 1-2 decades become more frequently used in urban hydrology along with increasing

length of recording period (Thorndahl et al., 2016), in particular for online applications (Pedersen et al., 2016). However, a few datasets now have sufficient lengths and with sufficient tracking of software and hardware changes to the recordings to allow construction of ground-truth recordings of more than 10 years of continuous recordings. Such series have shown to be useful in showcasing spatial variability of return levels or other extreme event characteristics at grid cells level (e.g. Goudenhoofdt et al., 2017; Panziera et al., 2016, 2018). However, few studies have studied extreme event characteristics over

the spatial extent of the events, and these are often limited to analysing area and intensity (Armon et al., 2020; Hamidi et al., 2017; Thorndahl et al., 2014). Here we apply a broad range of spatio-temporal characteristics in order to develop an automatic classification scheme of different event types and provide a better understanding of actual precipitation processes. A improved typology of extreme events may allow classification schemes that improve now-casting of precipitation as shown in e.g. Olsson et al. (2015) and regional models with better predictive capabilities as shown for Denmark in Madsen et al. (2017).

This study aims to quantify and describe spatial rainfall as a function of temporal and spatial dynamics, rainfall types and seasonal variation. A principal component analysis and clustering algorithm are applied to analyse selected event-descriptive variables and their internal correlation. The study describes rainfall extremes using spatio-temporal variables, in order to classify events according to meteorological origin and implication on urban drainage response.



## 2 Data and case area

### 2.1 Case area

The case area is a 38x48km rectangle (1824 km$^2$) including the catchment of the river Wupper in North Rhine-Westphalia, Germany. It stretches from the Rhine lowland in southwest to the more hilly area in east, with steep valleys around the river

Wupper. The elevation varies from 31 meters to 483 meters above sea level (see Figure 1). The mean annual precipitation in the area ranges from 770 mm to 1352 mm due to strong orographic effects with lowest precipitation in low lying areas and most precipitation in the highest elevated areas. Climatologically the area is close to the Atlantic Ocean. The prevailing weather conditions are warm and humid air coming from west into the Bergisches Land. The Bergisches Land is the first major barrier and causes orographic rainfall on the western side. In general the climate is mild with a wet and warm summer from May to

September. Summer precipitation is increasingly dominated by convective thunderstorms while winter precipitation are caused by frontal events from a western direction (DWD, 2018; Klima.org, 2018; LANUV, 2018). Due in part to high urbanisation (city of Wuppertal has approximately 350.000 inhabitants and the whole area about 900.000), small-scale but highly intense convective rainfall may cause flash floods with significant damage potential. This was demonstrated on 29 May, 01 and 09 of June 2018, where such storms caused substantial flooding e.g. in the city centre of Wuppertal (and other municipalities as

well) damaging buildings as well as significant infrastructure such as university buildings, gas stations, and a shopping centre. Nearly 900 action points from fire brigade, technical services and Wupperverband (The utility company of the area) were recorded and the reported damage sum in Wuppertal was about 1 Mio. € (the estimated total damage sum is about 7 Mio € for Wuppertal) (Wuppertal.de, 2018). Hence there is a recognised need for better understanding of spatio-temporal characteristics of extreme events in order to predict their occurrence and impact over the area.

### 2.2 Data

Radar data from the Deutsche Wetterdienst (DWD) Doppler C-band radar network was used in this study (5-minute temporal resolution, 1x1 km spatial resolution). The dataset comes from the Wupperverband and spans 13 years, from the 1$^{st}$ of November 2000 to the 1$^{st}$ of November 2013. The case area is within the range of the Essen radar and partly within the range of the Flechtdorf and Neuheilenbach radars (see Figure 1). The dataset is a weighted composition of the three radars (Einfalt

and Lobbrecht, 2011).

The dataset is post processed by hydro & meteo GmbH on behalf of the Wupperverband. Data is corrected with respect to blockage, clutter and attenuation. The reflectivity (Z) rainfall intensity (R) relationship is depended on the reflectivity, and can for convective rainfall be described as $Z = 256 \cdot R^{1.42}$. In the Wupperverband district the radar data is adjusted to rain gauge

data on a daily basis, with a correction factor per gauge in a 1 km correction grid using inverse distance weighting. Rain gauge data is beforehand visually inspected and compared to nearby gauges in order to secure the quality. There are 60 rain gauges within the area of the Wupperverband. The post processed dataset have less than 5 % difference from annual ground truth





(Frerk et al., 2012). The probability for a difference of more than 5 mm between the corrected dataset and independent stations (stations not used for radar data adjustment) is 1.4 cases per station per year. The probability of a difference above 10 mm is 0.1 cases per station per year (for the methodology, see Einfalt and Frerk, 2011). An analysis of the first 10 years of data shows that there is an underestimation of extreme events for short time steps, which is reduced for larger aggregation time steps. The

underestimation for a 10 km$^2$ pilot area were ranging from approximately 30% (5 minute time step) to 0% (daily time step) (Einfalt and Scheibel, 2015).

## 3 Methodology

### 3.1 Extreme events

Extreme events are identified based on time series dataset and defined based on a Peak Over Threshold method (Coles, 2001).
A Type II censoring is applied with a prefixed number of 39 extreme events, equal to an average of 3 events per year (Mikkelsen et al., 1995). Three types of extreme events are considered, 15-minute, 1-hour and 24-hour extreme events, based on the maximum average intensity for 15 minute, 1 hour or 24 hours respectively. The different temporal resolutions are selected to understand both convective (shorter temporal scale) and frontal (longer temporal scale) properties of extreme events. Convective events are high in intensity and often very local events develop in warm temperatures, whereas frontal events are
spatially large events with large event depth and duration but often lower intensities than convective events (Doswell et al., 2005). While 1-hour and 24-hour extreme events are commonly sampled to understand differences between convective and frontal activity, 15-minute extreme events are chosen to understand the difference between hourly and sub-hourly extremes, as datasets are often not accessible in sub-hourly resolution with sufficient record length.

### 3.2 Spatial dependence in sampling of extreme events

Typical extreme event definitions are based on time series of point data. To our knowledge there is no generally applied procedure to sample extreme events from multi-site or areal measurements such as radar data. Based on the time series dataset we examine four spatial scales to identify rain events in order to determine the number of grid cells that should be considered when selecting extreme events for further analyses. All methods identify the number of rain events in the data period, average length of rain events, average maximum number of grid cells registering each event, and seasonal distribution of rain events.
The methods are suggested to avoid a subjective selection of extreme events for analysis. Independent of the number of grid cells used in the methods, the sampled extreme events are all analysed based on the full gridded dataset of the case area. The method therefore only determines which events that are analysed. The sampling strategies (SS) are listed below in order of increasing number of grid cells used in the selection process:

    SS1.        Sampling from 1 grid cell
SS2.        Sampling from 5 grid cells (area of approximately 10x8km)
    SS3.        Sampling from half of the case area 38x24km (every ninth grid cell)



SS4.        Sampling from the entire case area 38x48km (every ninth cell)

The strategies correspond to using between one and 187 cells out of the 1824 cells when selecting the extreme events. The spatial extend of each of the sampling strategies are indicated in Figure 2.

*SS1, Sampling from 1 grid cell*

The simplest sampling strategy is choosing one grid cell from which rain events are identified. In this case a grid cell in the middle of the catchment is chosen. Rain events separated by dry periods less than 24 hours apart are aggregated to one event in accordance with (Madsen et al., 2002, 2009). Rainfall intensities below 1mm/hr is considered dry.

*SS2, Sampling from 5 grid cells*

The second sampling strategy considers 5 grid cells in a spatially small area on the same side on the mountain as the predominant westerly wind direction. The spatial area of the sampling method represents a small typical urban catchment, the locations of rain gauges in a city or the size of a single grid cell in most climate models. Precipitation series from the five grid cells are merged into a combined precipitation series. Precipitation occurs when at least one of the locations measures rainfall
and stops when all locations measures no rainfall (drizzle threshold of 1mm/hr). Events are aggregated using the same approach as when sampling from one grid cell (24 hour dry period).

*SS3, Sampling from half of the case area (every ninth grid cell)*

The third sampling strategy for rain events considers a larger part of the catchment. Here half of the catchment is considered,
with a total size of 38x24 km. Every ninth grid cell in this area is selected, yielding a selection of 104 grid cells. Precipitation events are defined using the same approach as when sampling from 5 grid cells.

*SS4, Sampling from the entire case area (every ninth grid cell)*

The fourth sampling strategy concerns the entire catchment. Every ninth grid cell is selected and precipitation time series are
merged with a 24-hour dry period between independent events. A total of 187 grid cells are considered.

### 3.3 Data analysis

### 3.3.1 Spatial variation

Extreme events from five independent grid cells are sampled with SS1 to clarify how the change of the grid cell used in SS1 impacts which extreme events are sampled. The grid cell from SS1 is used as a reference and compared to the four remaining
grid cells from SS2 (see Figure 2) to identify the small-scale variability in sampled extreme events. All four grid cells are approximately 5 km from the reference grid cell. The extreme events sampled with SS1 for the five grid cells are compared calculating the number of concurrent events using the method outlined in section 3.3.3.


### 3.3.2 Seasonal variation

The seasonal variation of occurrence of extreme events are analysed for 15-minute, 1-hour and 24-hour extreme events. Extreme events are sampled using SS1, but all of the five grid cells included in SS2 are analysed separately to test for local variations in the climate (see Figure 2). The analysis is based on four seasons: winter (December-February), spring (March-
May), summer (June-August) and fall (September-November).

### 3.3.3 Spatial correlation

The spatial correlation between 4950 pairs of grid cells (100 randomly selected grid cells) is calculated, applying the framework of spatial correlating structures by Mikkelsen et al. (1996). The spatial correlation is not calculated for all grid cells, as the correlation between neighbouring cells is very high and the benefit does not match the extra computational effort. The method
calculates the spatial correlation by estimating the correlation of extreme events that are meteorologically dependent. The unconditional correlation coefficient ρ between a pair of grid cells ($A$ and $B$) is calculated by identifying concurrent events. If it is assumed that the start ($t_s$) and end ($t_e$) times of all events are known, concurrence between the $i$'th event at grid cell $A$, $Z_{Ai}$ and the $j$'th event at grid cell $B$, $Z_{Bj}$ is defined as:

$$\{Z_{Ai}, Z_{Bj}\}: \left[ t_{si} - \tfrac{1}{2}\Delta t, t_{ei} + \tfrac{1}{2}\Delta t \right]_A \cap \left[ t_{sj} - \tfrac{1}{2}\Delta t, t_{ej} + \tfrac{1}{2}\Delta t \right]_B \neq \varnothing \tag{1}$$

where $\Delta t$ is a lag time introduced to ensure that events can be concurrent events though travelling time means that these events do not overlap in time. $\Delta t$ was in this study set to 11 hours equal to the $\Delta t$ used in Gregersen et al. (2013). Based on the sample of concurrent events and the sample of not concurrent events in a pair of grid cells, the unconditional covariance is estimated as:

$$Cov\{Z_A, Z_B\} = Cov\{E\{Z_A|U\}, E\{Z_B|U\}\} + E\{Cov\{Z_A, Z_B|U\}\} \tag{2}$$

where $U$ is a stochastic variable which has the value of 1 for concurrent events and otherwise 0.

Given by the definition of $Cov\{E\{Z_A|U\}, E\{Z_B|U\}\}$ and $E\{Cov\{Z_A, Z_B|U\}\}$ in Mikkelsen et al. (1996), the unconditional correlation coefficient $\rho$ can now be estimated by dividing the unconditional covariance with the standard deviation for the two grid cells. Following the procedure proposed by Gregersen et al. (2013) the data is hereafter divided into bins based on distance between stations and the average $\rho$ for each bin is calculated in order to minimise noise in the data set. An exponential
function is fitted to data, relating the distance between a pair of grid cells with the unconditional correlation coefficient $\rho$. The e-folding distance is then found as the distance where the unconditional correlation have decreased to 1/e, based on the fitted exponential function (Gregersen et al., 2013).

### 3.4 Characterisation of events

The three sets of 39 extreme events are characterised by 17 variables chosen to describe a variety of event properties (see Table
1); these can be further aggregated into six categories: *Duration*, *intensity*, *wet area coverage*, *depth*, *rain cell properties* and *movement*. Rain cell properties and movement are described with a simple rain cell identification and tracking algorithm as





described below. The analysis to characterise the events are done considering the entire case area, analysing the gridded dataset in 5-min resolution for all timesteps covered by in the event.

### 3.4.1 Rain cell identification

Rain cells are identified in each time step by assigning an intensity threshold and an areal threshold. The intensity threshold is
set to 25% of the maximum 5-minute intensity for the given event with a minimum threshold of 7 mm h[-1]. The areal threshold is set to a minimum coverage of 10 km$^2$. The thresholds are set to align with well-known cell identification and tracking algorithms (Dixon and Wiener, 1993; Handwerker, 2002; Kyznarová and Novák, 2009; Peleg and Morin, 2012). The intensity threshold is chosen to be event varying to distinguish between different rain cell types (e.g. convective and front cells) and secure a high threshold for all events which result in a more stable tracking of a clear cell centre (Dixon and Wiener, 1993).
Rain cells with an area below the areal threshold are disregarded to avoid noise in the overall tracking from multiple small cells (Dixon and Wiener, 1993). An ellipse is fitted to each of the identified rain cells, with the coordinates for the centroid, length of the axis and orientation in degrees between major axis and east-axis (Belachsen et al., 2017; Peleg and Morin, 2012).

### 3.4.2 Rain cell tracking

Various complex rain cell tracking algorithms can be found in the literature (Dixon and Wiener, 1993; Handwerker, 2002;
Kyznarová and Novák, 2009). For describing the overall moving direction and velocity of each rain event this study has developed a simple tracking algorithm. Rain cell movement is recorded by linking the identified rain cells in each time step together in a simple tracking algorithm based on the position of the centroid. Tracking is based on the moving direction and velocity from last time step, which is used to predict the approximate position of the rain cell in the next time step. The rain cell with the centroid closest to the predicted position of the rain cells centroid is linked to the rain cell in the previous time
step with no further evaluation of the fit. A maximum distance of 7.5 km, corresponding to a moving velocity of 25 m s[-1], from the predicted position of the rain cell to the linked rain cell is applied. For new rain cells, the position of the rain cell in time step one is the predicted position of the rain cell in next time step. The tracking algorithm manages birth, tracking and death of rain cells. If splitting of a rain cell occurs, the algorithm will treat it as continuous tracking of the rain cell and a birth of a new rain cell. In case of merging of two rain cells, the algorithm will classify it a death of one rain cell and continue tracking
of the other rain cell.

### 3.5 Statistical analyses

All statistical analyses are performed using normalised data, i.e. a transformation to ensure mean zero and variance one. All analyses are carried out in R using the build-in R Stats Package version 3.4.1 and cluster package 2.0.7-1 (Maechler et al., 2019).





### 3.5.1 Principal component analysis

Principal Component Analysis (PCA) is used to determine the number of dimensions necessary to describe a dataset; here the number of the 17 variables used to characterize the events as described in Section 3.4 Characterisation of events. The eigenvector with the $i^{th}$ largest eigenvalue ($\lambda_i$) is noted the $i^{th}$ principal axis, where $PC_i$ represent the projection of data on the $i^{th}$ principal axis (Morrison, 1967). The percentage of the variance, which $PC_i$ describes, is calculated as the percentage of the sum of the eigenvalues based on the $i^{th}$ eigenvalue (Morrison, 1967).

Two tests are applied to determine the number of dimensions necessary to describe data. The first test is an approximate test to estimate the number of significant PC's based on the magnitude of the eigenvalues. The hypothesis tested is that the last $k+1$ to $m$ eigenvalues are similar and therefore non-significant, where $m$ is the total number of eigenvalues. The test is described in Lawley and Maxwell (1963) and Anderson (1984) as:

$$H_0: \lambda_1 \geq \cdots \geq \lambda_k \geq \lambda_{k+1} = \cdots = \lambda_m \tag{3}$$

The test statistic is defined as:

$$z_2 = -n * \ln\left(\frac{\prod_{i=k+1}^{m} \lambda_i}{\hat{\lambda}^{m-k}}\right) \tag{4}$$

where $\hat{\lambda}$ is defined as:

$$\hat{\lambda} = \sum_{i=k+1}^{m} \lambda / (m - k) \tag{5}$$

The second test estimates the number of effective spatial degrees of freedom based on the eigenvalues and was proposed by Bretherton et al., (1999) as:

$$N_{eff} = \frac{\left(\sum_{i=1}^{m} \lambda_i\right)^2}{\sum_{i=1}^{m} \lambda_i^2} \tag{6}$$

### 3.5.2 Cluster analysis

Clustering is performed on the dataset to identify similarities between the events based on all variables. The K-means clustering algorithm presented by Hartigan (1975) and Hartigan and Wong (1979) is selected as partitioning clustering method. If $l(i)$ describes the cluster where the event $i$ is contained and $l$ represents any cluster then $D[i,l(i)]$ denotes the Euclidean distance between event $i$ and cluster centre $l(i)$ and similarly $D[i,l]$ denotes the Euclidean distance between event $i$ and the centre of any other cluster. Reallocation of events to another cluster is done if it decreases the error.



## 4 Results

### 4.1 Spatial dependence in sampling of extreme events

There is a relatively small decrease in number of events when increasing the number of grid cells considered in the sampling strategy. The largest decrease in number of events is found between SS1 (1 grid cells) and SS2 (5 grid cells), while a smaller

decrease is found between SS2 and SS3/SS4 (half case area/full case area) (Table 2). In this study a drizzle threshold of 1 mm/hr is applied, considering intensities below this threshold dry. If this threshold was not applied, and all grid cells considered should be completely without rainfall to separate events, a quite distinct reduction in number of events would be the case between SS2 and SS3/SS4. Not applying the drizzle threshold and considering more grid cells would result in meteorological independent events to be merged. The average event length increases with increasing number of grid cells considered, as a

larger period of the time where an event moves over the case area is detected. Events which are merged when considering more grid cells are especially summer events, as the proportion of summer events is most affected by the increase in considered grid cells (Table 2). The relatively small decrease in number of events, when considering an increasing number of grid cells, indicates that in a case area of this size all grid cells detect almost the same pool of events. From this, using SS1 for sampling extreme events seems valid for a case area of this size. A further advantage with SS1 is then the ability to use the theories for

extreme precipitation developed for point measurements. The disadvantage of using SS1 as sampling strategy is that the method cannot identify the entire duration an event moves through the case area and that the true peak intensity of the event is often not found in the sampling point used to select the events. In order to accommodate for this a sampling strategy considering the entire case area must be used. To consider a larger area when sampling extreme events gridded data is needed. Here a spatial definition can be used to outline events, e.g. as done in various tracking algorithms, but this method gives a very

different event definition from the one used for rain gauge data. In order to use the knowledge about extreme events from rain gauge data and be able to compare the results obtained to studies using rain gauge data, SS1 is chosen as the sampling strategy for this study.

### 4.2 Data analysis

#### 4.2.1 Spatial variation

The spatial variation in which extreme events are sampled are largest for 15-minute extremes and smallest for 24-hour extremes (see Table 3). When using SS1 the number of concurrent events between the reference cell (black, Figure 2) and the four surrounding grid cells (grey filled, Figure 2) increases with increasing sampling duration. Only approximately 47% for the 15-minute events and 55% of the 1-hour extreme events are the same events for the four surrounding grid cells when comparing to the reference grid cell, while 80% of the 24-hour extreme events are the same (Table 3). This indicates a more localised

spatial extent with lower sampling duration. The spatial variation shows that the localised structure of 15-min and 1-hour events results in a large differences in the sampled extreme events. A lower threshold could result in larger similarities in sampled extreme events between grid cells as the pool of event sampled from is very similar for all grid cells (see section 4.1



Spatial dependence in sampling of extreme events). It is believed that the most severe extreme events in the case area is sampled for all grid cells, even though the ranking could be different between the grid cells. While the pool of events to select extreme events from is very similar for the five analysed grid cells (see section 4.1 Spatial dependence in sampling of extreme events) the selected extreme events varies depended on how localised the sampled extreme events are. Samplings strategies

were suggested based on the time series dataset for a better comparison with rain gauge data and SS1 was selected for further analysis as the best of the proposed sampling strategies. Still it is clear from the spatial variation that the selected grid cell has a large impact on which extreme event is sampled, increasing with decreasing event duration. Despite SS1 being the best sampling strategy using the method and statistics from rain gauge data, this strategy does not sample all events in the case area with intensities above a certain threshold and might not have the right composition of types of extreme events. Another

approach could be to sample extreme events from each grid cells in the case area, as shown in (Goudenhoofdt et al., 2017; Panziera et al., 2018), but these papers do not propose a spatial definition of independent events and merging of non-independent events in order to analyse spatio-temporal characteristics of events as proposed in this article. Other articles have suggested different methods to sample extreme events from gridded data, with no methods being similar and with very different definition of extremes (Armon et al., 2020; Hamidi et al., 2017; Panziera et al., 2016; Thorndahl et al., 2014). Common for the

proposed event sampling strategies in these articles is a difficulty in defining the beginning and end of events, and no methodology to define a suitable number of events sampled or the extremity of the sampled events. Many papers have used various ways of event tracking (e.g. Denoeux et al., 1991; Kyznarová and Novák, 2009; Peleg and Morin, 2012). Tracking algorithms gives a clear definition of beginning and end of events, but lacks a methodology for extremity and suitable number of events to sample. This indicates that more work is needed in defining a methodology for sampling extreme events with a

common sampling strategy and definition of extremity as known from rain gauge data in terms of Annual Maximum, Peak-over-threshold and return periods.

### 4.2.2 Seasonal variation

The seasonal variation of occurrence of extreme events for each of the five grid cells used in SS2 can be seen in Figure 3. There is very little variation between the five grid cells in seasonal variation in occurrence of sampled extreme events. Between

types of extreme events the difference in seasonal occurrence of extreme events is largest between 1-hour and 24-hour extreme events, and quite similar between 15-minute and 1-hour extreme events. 15-minute and 1-hour extreme events almost only occur in the summer while 24-hour extreme events are more uniformly distributed over the year. This corresponds well with the seasonal difference in precipitation in the area (ExUS, 2010; Quirmbach et al., 2012) and the expectance of differences in seasonal variation between different event types, convective vs. front events as shown in e.g. (Gregersen et al., 2013).

### 4.2.3 Spatial correlation

The spatial correlations calculated between 4590 pairs of grid cells for 15-minute, 1-hour and 24-hour extreme events are shown in Figure 4. The spatial correlation decreases with increasing separation distances between pairs of grid cells. The



spatial correlation for 15-minute and 1-hour extreme events decreases faster with distance than for the 24-hour extreme events and the 15-minute extreme events faster than the 1-hour extreme events. This indicates that 15-minute and 1-hour extreme events are more localised and small-structured events while the 24-hour extreme events are spatially larger events. This corresponds well with the results from Peleg et al. (2013). From the fitted exponential functions, the e-folding distances are

calculated to be 5.7, 9.0 and 22.2 km for 15-min, 1-hour and 24-hour extreme events, respectively. Several studies have calculated the spatial correlation based on rain gauge data from a dense network of rain gauges around the world (e.g. Gregersen et al., 2013; Mandapaka and Qin, 2013; Peleg et al., 2013; Villarini et al., 2008). The reported results vary as a function of the local conditions, including climatic area, distances between rain gauges, spatial extent, sample size etc. Besides varying conditions these studies show 1-hour e-folding distances which are well aligned with the 1-hour e-folding distance in this

study, except for one study (Villarini et al., 2008) which in general reported longer e-folding distances. The 24-hour e-folding distance in this study is in general shorter than what is found in other studies. Many of the studies have a small study area and few data points with long distances between. This increases the uncertainty of the estimated values, which could be the reason for the difference between the e-folding distance in this and other studies. The e-folding distance for 24-hour extreme events in this study is calculated based on grid points where large bins of data even for long separation distances are available. The

e-folding distance of 22.2 km is less than half of the spatially extent of the case area. Another reasoning for shorter e-folding distances in this study could be that it focuses only on extreme events, while other studies calculates the spatial correlation for all events. Gregersen et al. (2013) is the only other study focusing solely on extreme events. Here the 1-hour e-folding distance is shorter (5 km vs. 9.0 km) and the 24-hour e-folding distance longer (37 km vs. 22.2 km) compared to this study, but the results show similar orders of magnitude. The differences between the study of Gregersen et al. (2013) and this study could

perhaps be assigned to the limitations of using rain gauges or differences between a predominantly coastal climate and a more continental climate. Two studies identify spatial correlation for sub-hourly extremes; Peleg et al. (2013) calculate correlation distances of 6 and 9 km for 10-minute and 30-minute events respectively, while Villarini et al. (2008) get a correlation distance of approximately 20km for 15-minute events.

### 4.3 Event characterisation

Using sampling strategy SS1 15-minute, 1-hour and 24-hour extreme events are sampled. The sampled extreme events are described by the chosen 17 variables in the event analysis using the dataset within the case area illustrated in Figure 1 (left). The data for all sampled extreme events are shown in the supplementary material (Table A, Table B and Table C). For the 39 sampled extreme events for each of the three event sampling durations, the 15-min events consists of 28 summer events (and 11 non-summer events), the 1-hour event of 27 summer events and the 24-hour events of only 9 summer events. Differences

between variables describing 1-hour and 24-hour extreme events are in particular pronounced for the variables *Duration*, *Maximum 15 minute intensity* and *Maximum depth*, which can be related to the differences between convective events and events within frontal systems. Differences between 15-minute and 1-hour extremes are small when comparing the 17 chosen variables which can be explained by the large overlap of 27 events sampled as both 15-min and 1-hour events. Difference





between 15-minute and 1-hour events are found in variables such as *Ratio 15min* and *Ratio depth* which indicates a larger variability within 15-minute events than 1-hour events. Ten events is sampled as both 15-minute, 1-hour and 24-hour extreme events; these are listed in Table 4. The results from the event characterisation, in relation with the results from the seasonal variation and spatial correlation indicate that the events sampled are representative for extreme events over the year in the area.

The study focuses on properties of precipitation only and hence does not include e.g. temperature or atmospheric pressure. Such variables might be relevant as shown by (e.g. Lochbihler et al., 2017; Peleg et al., 2018), in particular if the purpose is to do simulations with a weather generator by conditioning it by the current state of the atmosphere.

### 4.4 Statistical analyses

#### 4.4.1 Principal component analysis

The PCA is calculated using the aggregated dataset of 15-minute, 1-hour and 24-hour extreme events. The combined dataset is normalised before the calculation of the PCA. In Table 5 the weighted composition of variables in each of the first nine *PC*s can be seen. $PC_1$ and $PC_2$ are influenced by most of the variables describing the means of the events. $PC_1$ is positively influenced by variables such as *Duration, Max depth* and *Mean cell lifetime* and negatively influenced by *Max 15min intensity, Max 1hr intensity, Ratio 15min* and *number of cells*. $PC_2$ is mostly influenced by *Max 24hr intensity* and the depth variables.

$PC_3$ can be summarised as the interaction between movement and spatial extent, positively influenced by *Duration,* and negatively by *Mean velocity, Mean wet area* and *Max cell lifetime. PC_4* describes the movement of the rain cells in the events and is mostly influenced by *Standard deviation of direction* and *Mean direction.* The two first *PC*s explain 53% of the total variance and the first nine *PC*s should be considered if 95% of the variance must be explained. Based on the eigenvalues 14 *PC*s are significant when using the approximate test in Eq. (3-5). The alternative test suggests that there are 5.6 effective *PC*s.

As such, five to nine dimensions should be, and up to 14 dimensions could be, considered in order to describe the variability of the events when considering both 1-hour and 24-hour extreme events.

When projecting the 15-minute, 1-hour and 24-hour events into the two first *PC*s a clear clustering can be seen, with 15-minute and 1-hour extreme events as one cluster and 24-hour extreme events as another cluster, while there is no distinct difference

between 15-minute and 1-hour extreme events (Figure 5, left). The distinction between the two clusters is determined by the $PC_1$, where the 24-hour events have larger positive values than the 15-minute and 1-hour events. This indicates that the observed differences between convection dominated (15-minute and 1-hour) and front system (24-hour) extreme events can be described by scaling across the variables important for $PC_1$. $PC_2$ show a scaling in extremity with increasing value. When the combined dataset of extreme events is projected into $PC_1$ and $PC_3$, four 24-hour events are distinct from the rest of the

events with very negative $PC_3$ values. These events have short durations and large mean wet area as well as many rain cells and long maximum cell lifetime, together resulting in a negative $PC_3$ value.





The seasonal variation of the sampled extreme events visualised by the two first *PC*s can be seen in Figure 6. A distinction between summer and non-summer events is seen, reflecting the difference in the seasonal variation between convection dominated (15-minute and 1-hour) and front system (24-hour) extreme events. From the results of the PCA it is clear that $PC_1$ describes the difference between the two main types of extreme events, $PC_2$ the extremity and that $PC_3$ and onwards mostly

describes very specific features about few events (Table 5). It is furthermore seen that all variables helps describing the variability in the events and none are insignificant.

### 4.4.2 Cluster analysis

The K-means clustering algorithm is performed with a predefined number of two and four clusters based on the outcome of the PCA (Figure 7). Two clusters was selected to evaluate if the selected variables was able to distinguish between convective

and front system events. The two clusters show a distinction between 15-minute and 1-hour events in one cluster and 24-hour extreme events (Figure 7 left). The first cluster primarily consist of 24-hour extreme events with few 15-minute and 1-hour extreme events and opposite in the second cluster. Dividing the data into three clusters separated the two most extreme events (top left corner in Figure 7) into a separate cluster, with the two other clusters similar to the clusters defined with a predefined number of two cluster (results not shown).

When selecting four clusters, Cluster 1 is defined by very high *maximum 15-minute*, *1-hour* and *24-hour intensities* and large *maximum depth*, and may be characterised as convective activity within extreme frontal events (Figure 8). Cluster 1 consists of 2 known very extreme events on the dates; 19/06-2013 and 06/08-2007 (Figure 7 right). Cluster 2, consists mostly of 24-hour events with characteristics such as long *Duration*, low *Ratio between maximum and mean depth* and low *Maximum 1*

*hour intensity* (Figure 8). These events can be classified as extreme frontal events with little or no convective activity. Few 15-minute and 1-hour extreme events are within this cluster; the ones present differ from the rest of the 15-minute and 1-hour extreme events by long *Duration*, and large *Minimum* and *Mean depths*. Cluster 3 consists of four 24-hour events which was separated on $PC_3$ (see Figure 5, right) and are characterised by having low depths and large cell numbers (Figure 8). The events in Cluster 3 can be considered events that are not so pronounced extreme; i.e. events that are sampled as extreme but should

not pose a problem with respect to flooding. The last cluster, Cluster 4, consists of 15-minute and 1-hour events and few 24-hour extreme events and events sampled as in all sampling durations (Figure 7 right). Cluster 4 contains 77% of the sampled 15-minute extremes and 74% of the sampled 1-hour extremes and are clearly convective extreme events with high intensities and low depths.

Structures of clusters from 1 (all events in same cluster) to 117 (all events in separate clusters) was assessed by a hierarchical clustering with two methods (ward and average linkage), showing same overall clustering structures, though with minor differences between methods (see supplementary material Figure A). The hierarchical clustering was used to assess the possible number of clusters and also testing if a larger number of clusters contributed with more information. The analysis





supported both the amount of clusters analysed as well as the interpretation of the distinction between clusters obtained in the K-means clustering analysis.

The results show that the combination of PCA and cluster analysis can be used to distinguish between convective, frontal and mixed events (Clusters 1, 4 and 2 respectively) when enough describing variables are considered to characterise the events in radar data. For the applicability a number of four clusters was found reasonable, yet these clusters was fitted using all 17 *PC*s and variables, as several *PC*s was found significant and no variables was disregarded.

## 5 Conclusion

The quantitative methods employed are able to distinguish between types of rainfall based on measured high-resolution
extreme spatial rainfall. They are in line with the results from previous works where the understanding of spatial rainfall is described qualitatively. The quantitative analysis was possible due to the high quality high resolution data set of merged quality checked rain gauge and quality checked radar data. The virtues of highly accurate extreme intensities are thus combined with good spatial coverage which makes the results more credible than analysing radar data or gauge data only. The seasonal variation and spatial correlation of the analysed extreme events confirm a clear difference between 15-minute/1-hour extreme
events and 24-hour extreme events which can be described as a distinction between convective and frontal events. The differences between 15-minute and 1-hour are however less pronounced than expected, both when considering the event selection and in the subsequent analyses of the spatio-temporal characteristics. Four sampling strategies for sampling spatial extreme events were analysed and it was found that it was necessary to sample extremes using a threshold for a small region in order to avoid long events with sub-events that are meteorologically independent. The analysis of spatial correlation showed
that 15-minute and 1-hour extreme events are very local. It was shown that a least 50 % of sampled extreme events would change if another grid cell within a radius of approximately 5 km were chosen as sample point, with decreasing similarity in sampled extreme events with increasing distance between the compared grid cells. This study suggests that further development on a sampling strategy for sampling spatial extreme events is needed.

Events were characterised by 17 variables giving a thorough description of the spatio-temporal variability of the events. All variables contribute with information about the analysed extreme events, even though there are correlations suggesting that not all dimensions are necessary. The PCA suggests five to nine dimensions necessary to describe the data, but up to 14 *PC*s were found significant, implying that most variables are relevant to consider. From the PCA and cluster analysis it was possible to distinguish between four different storm types: convective, frontal, mixed very extreme and borderline extreme events.
Using more than four clusters made it difficult to interpret the results in a simple and applicable manner and appeared to mainly be due to overfitting of data rather than contribute to an overall understanding of spatial precipitation properties.



A simple rain cell identification and tracking algorithm was developed for the study to describe the overall tendency in rain cell lifetime, number, direction and velocity of extreme events. For the purpose of this study the relatively simple algorithm proved to be sufficient to give a realistic description of the related variables.

5    The study contributes to the discussion on good practises of automatically identifying, defining and analysing single storm events in radar rainfall data sets. The methodology can be used to objectively classify rainfall events in spatial radar data based on measurable variables and, thus, act as a data filter for determining rainfall events of hydraulic and urban drainage interest.

**Acknowledgement**

The radar data used is a quality-controlled composite product from hydro & meteo GmbH based on the polar data product from the Deutscher Wetterdienst (DWD) radar network. Data is owned by the Wupperverband and has been made freely
10    available for this research purpose. Inquiries regarding the data should be addressed to the Wupperverband.



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



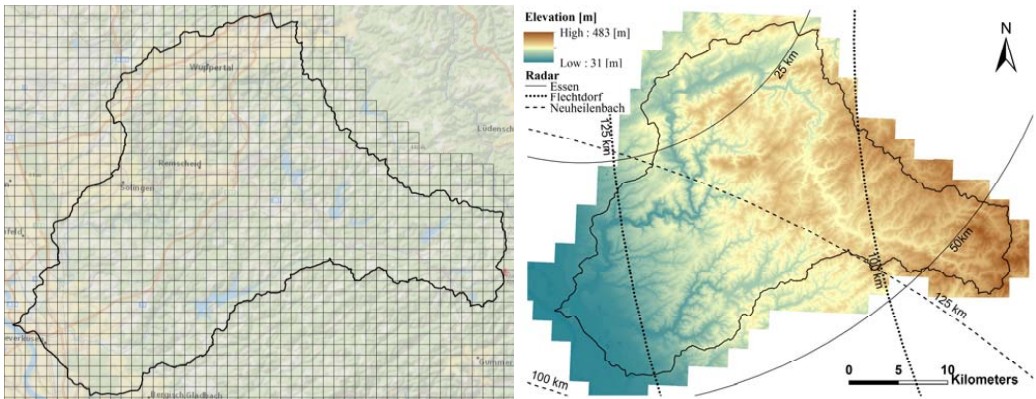

**Figure 1: Overview of the case area. Left: Gridded area represents the part of the catchment where time series data is produced. Right: Elevation in the Wupper catchment and distances from the three radars (Essen, Flechtdorf and Neuheilenbach) in the area.**

**Figure 2: Overview of the four sampling strategies: - SS1: Filled black cell, SS2: Black and grey filled cells, SS3: cells outlined in light grey and SS4: cells outlined in light and dark grey.**



**Table 1: Description of variables**

| Category | Variable | Unit | Description |
|---|---|---|---|
| Duration | Duration | Hours | From start to end with an extension of 2 hours in each end to consider the event in the entire case area. |
| Intensity | Max 15min | mm hr$^{-1}$ | Maximum average intensity for 15 minutes. |
| | Ratio 15min | - | Ratio between max 15 minute and mean 15 minute intensity. |
| | Max 1h | mm hr$^{-1}$ | Maximum average intensity for 1 hour. |
| | Max 24h | mm hr$^{-1}$ | Maximum average intensity for 24 hours. |
| Wet Area | Mean wet A | - | Average ratio of grid cells with precipitation above 1 mm/hr (wet cells) from each time step of the event. |
| Depth | Min depth | mm | Value of the grid cell with the lowest accumulated depth over the event duration in the case area. |
| | Max depth | mm | Value of the grid cell with the highest accumulated depth over the event duration in the case area. |
| | Mean depth | mm | Average depth considering all cells with a depth above 1 mm in the case area. |
| | Ratio depth | - | Ratio between max depth and mean depth. |
| Rain cell properties | Cell num | - | Number of tracked rain cell in the rain event. |
| | Cell life | hours | Average lifetime of the rain cells in the event. |
| | Cell life max | hours | Maximum lifetime of the longest living cell in the event |
| Movement | Mean vel | m s$^{-1}$ | Mean rain cell velocity |
| | Sd vel | m s$^{-1}$ | Standard deviation of velocity. |
| | Mean dir | Degrees | Mean moving direction of rain cells, compass degrees. |
| | Sd dir | Degrees | Standard deviation of direction. |

**Table 2: Results from the four sampling strategies described in Sect. 3.2**

| Sampling strategy | Number of events total | Average event length [h] | Average number of grid cells | Proportion of events in winter | Proportion of events in spring | Proportion of events in summer | Proportion of events in fall |
|---|---|---|---|---|---|---|---|
| SS1 (1 grid cell) | 982 | 28.65 | 1.00 | 0.22 | 0.23 | 0.30 | 0.24 |
| SS2 (5 grid cells) | 942 | 37.49 | 4.21 | 0.23 | 0.24 | 0.29 | 0.25 |
| SS3 (half case area) | 938 | 52.89 | 61.33 | 0.23 | 0.26 | 0.26 | 0.24 |
| SS4 (total catchment) | 933 | 57.48 | 99.26 | 0.23 | 0.27 | 0.26 | 0.24 |





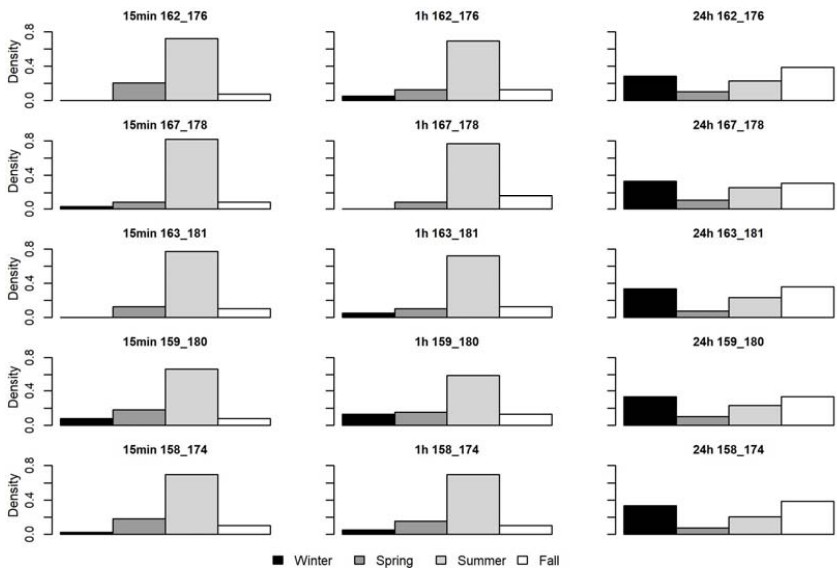

**Figure 3: Seasonal variation in occurrence of extreme events for each of the five grid cells filled in Figure 2.**

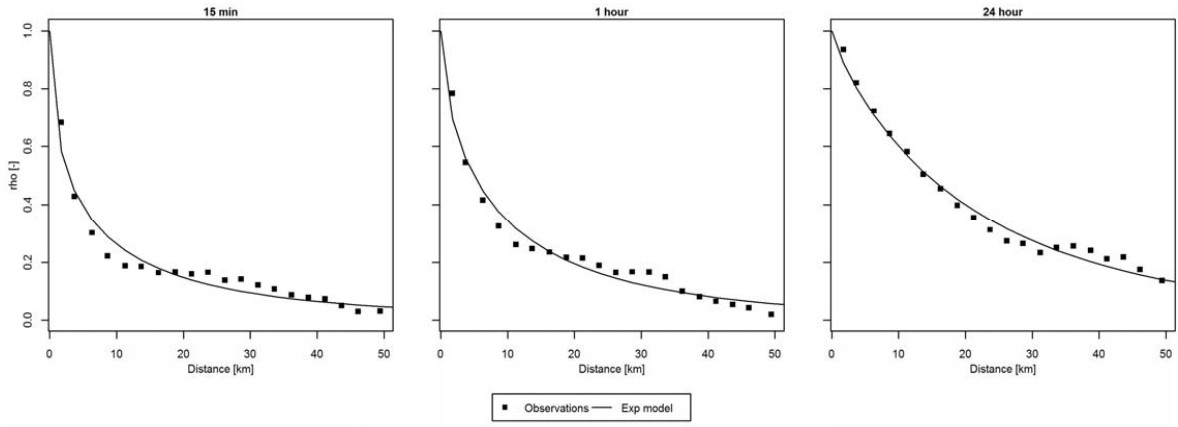

**Figure 4: Spatial correlation calculated for binned data of 100 grid cells, 4950 pairs.**



**Table 3: Comparison of extreme events sampled using SS1 for each of the five grid cells individually that is used in SS2. The grid cell from SS1 (black filled in Figure 2) is used as reference.**

| Name | 162_176 | 167_178 | 163_181 | 159_180 | 158_174 |
|---|---|---|---|---|---|
| distance (S,E) [km] | (0,0) | (-5,2) | (-1,5) | (3,4) | (4,-2) |
| distance [km] | 0.0 | 5.4 | 5.1 | 5.0 | 4.5 |
| Similar 15-min | 39 | 15 (38%) | 19 (49%) | 18 (46%) | 22 (56%) |
| Similar 1-hour | 39 | 19 (49%) | 21 (54%) | 23 (59%) | 22 (56%) |
| Similar 24-hour | 39 | 31 (79%) | 32 (82%) | 30 (77%) | 32 (83%) |

10  **Table 4: Overview of the 10 events sampled as all types of extreme events (15-min, 1-hour and 24-hour extreme events). The numbers refer to the event numbers in the supplementary material.**

| Date (LT) | 15-min | 1-hour | 24-hour |
|---|---|---|---|
| 17-07-2001 | 4 | 3 | 2 |
| 19-08-2002 | 7 | 6 | 6 |
| 05-10-2002 | 8 | 7 | 7 |
| 10-09-2004 | 13 | 12 | 14 |
| 29-06-2005 | 16 | 16 | 19 |
| 06-08-2007 | 21 | 22 | 25 |
| 10-08-2010 | 26 | 29 | 30 |
| 19-06-2013 | 36 | 36 | 37 |
| 22-07-2013 | 37 | 37 | 38 |
| 06-09-2013 | 39 | 39 | 39 |

…





Table 5: Composition of variables for the first nine PC's in a combined PCA including both 15-min, 1-hour and 24-hour extreme events.

| | PC1 | PC2 | PC3 | PC4 | PC5 | PC6 | PC7 | PC8 | PC9 |
|---|---|---|---|---|---|---|---|---|---|
| **Duration** | 0.21 | 0.28 | 0.32 | 0.11 | -0.14 | -0.28 | 0.14 | -0.40 | 0.04 |
| **Max 15min intensity** | -0.33 | 0.28 | -0.01 | 0.08 | 0.24 | 0.05 | 0.03 | -0.09 | 0.19 |
| **Ratio 15min** | -0.32 | 0.24 | 0.09 | 0.14 | 0.16 | 0.00 | 0.14 | -0.30 | 0.43 |
| **Max 1hr intensity** | -0.34 | 0.28 | -0.11 | 0.04 | 0.20 | -0.09 | 0.10 | -0.01 | -0.02 |
| **Max 24hr intensity** | -0.12 | 0.38 | -0.22 | -0.05 | 0.14 | -0.07 | -0.11 | 0.46 | -0.13 |
| **Mean wet area** | 0.22 | 0.11 | -0.42 | -0.12 | -0.08 | 0.37 | 0.04 | 0.25 | 0.48 |
| **Min depth** | 0.13 | 0.11 | 0.11 | -0.41 | 0.68 | 0.02 | -0.23 | -0.08 | -0.17 |
| **Max depth** | 0.28 | 0.35 | 0.10 | -0.01 | -0.21 | 0.02 | -0.03 | -0.01 | -0.05 |
| **Mean depth** | 0.10 | 0.47 | 0.01 | -0.04 | -0.12 | -0.14 | 0.04 | 0.16 | -0.04 |
| **Ratio depth** | 0.26 | 0.39 | 0.08 | 0.00 | -0.18 | -0.03 | -0.03 | 0.09 | -0.05 |
| **Number of cells** | -0.34 | -0.08 | -0.12 | 0.17 | -0.18 | -0.35 | 0.19 | 0.26 | 0.05 |
| **Mean cell lifetime** | 0.28 | -0.01 | -0.31 | 0.18 | 0.12 | -0.40 | -0.29 | -0.13 | 0.27 |
| **Max cell lifetime** | -0.15 | 0.16 | -0.35 | -0.32 | -0.22 | 0.27 | 0.35 | -0.37 | -0.36 |
| **Mean velocity** | 0.12 | -0.02 | -0.57 | -0.01 | 0.01 | -0.16 | -0.11 | -0.41 | 0.03 |
| **Sd velocity** | 0.31 | -0.03 | 0.11 | 0.18 | 0.27 | 0.31 | 0.50 | 0.04 | 0.25 |
| **Mean direction** | 0.24 | -0.04 | -0.22 | 0.39 | 0.33 | -0.20 | 0.45 | 0.12 | -0.40 |
| **Sd direction** | -0.07 | 0.13 | -0.05 | 0.65 | 0.00 | 0.48 | -0.41 | -0.12 | -0.26 |
| **Proportion of variance** | 30% | 23% | 13% | 8% | 7% | 4% | 4% | 4% | 2% |
| **Prop. of variance cumulative** | 30% | 53% | 66% | 74% | 81% | 85% | 89% | 93% | 95% |



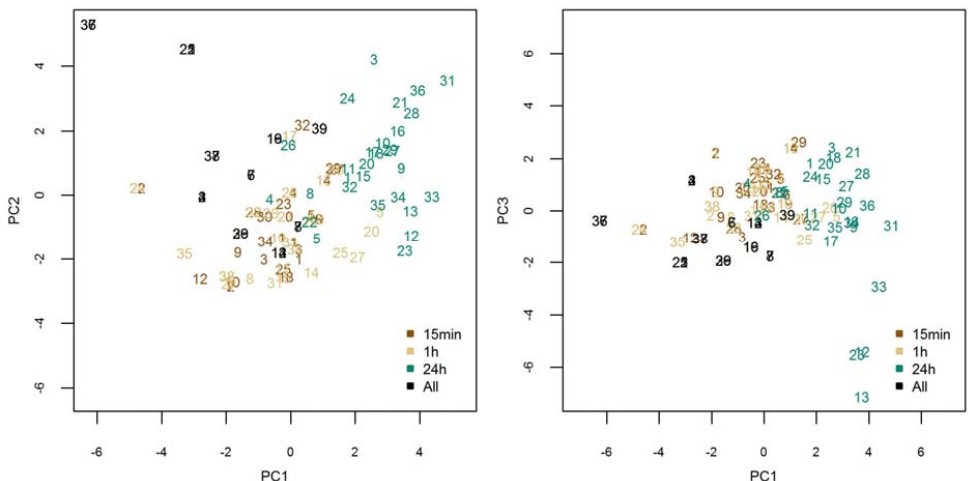

**Figure 5: Principal component analysis performed on 15-minute, 1-hour and 24-hour extreme events treated as a combined**
5  **dataset. Numbers refer to the event numbers in the supplementary material. Events sampled as all types of extreme events (both as**
   **15-minute, 1-hour and 24-hour extreme events) are marked in black. 15-minute extreme events are marked in dark brown, 1-hour**
   **extreme events in light brown and 24-hour extreme events are marked in green. Left: Projection into Principal Component 1**
   **(PC1) and PC2. Right: Projection into PC1 and PC3.**

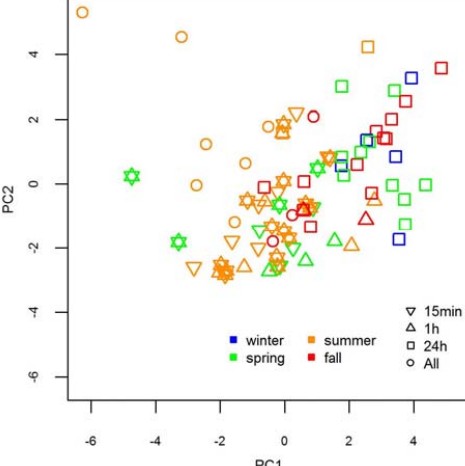

10  **Figure 6: Projection of extreme events into the two first PC's for the combined dataset. Colours indicate season (winter, spring,**
   **summer and fall) and shape indicate extreme event type.**

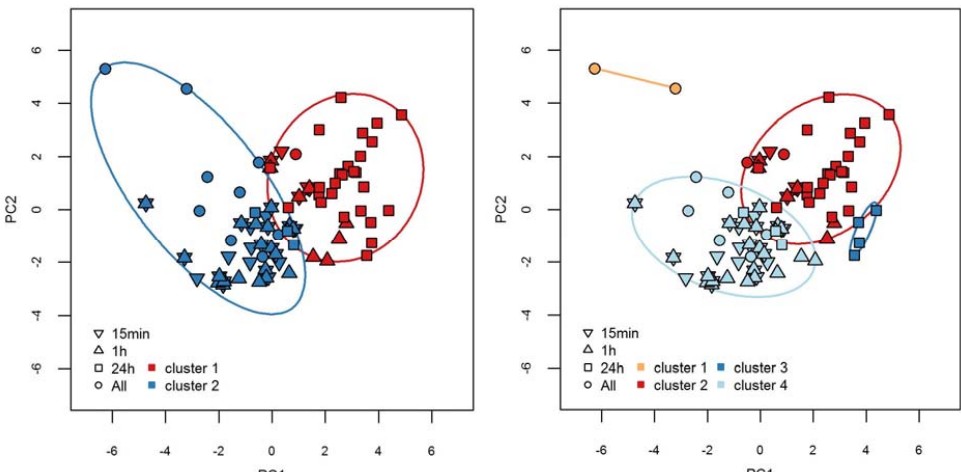

**Figure 7: K-means cluster analysis performed on the combined dataset for 15-min, 1-hour and 24-hour data. Left: Pre-defined number of two clusters. Right: Pre-defined number of four clusters.**

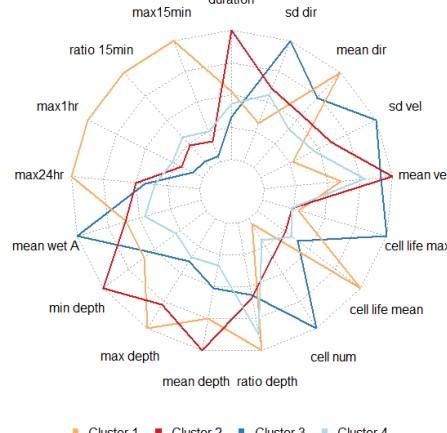

**Figure 8: Cluster composition of variables. Mean value for each variable over the events with each cluster. The mean values are normalised by the maximum mean value across clusters (scale between 0-1). Cluster numbers and colours refer to the clusters in Figure 7.**

