# Peer review of "Data-driven distinction between convective, frontal and mixed extreme rainfall events in radar data"

_Hydrology and Earth System Sciences, 2020_

## Referee Comment (RC1) · Anonymous Referee #1 · 11 Oct 2020

The paper "Data-driven distinction between convective, frontal and mixed extreme rainfall events in radar data" by Emma Dybro Thomassen et al. is in general well written, and it deals with an interesting and relevant topic for HESS. However, there are major issues to be addressed, as detailed below.

**Major comments:**

The main difficulty with this study is that it is not clear what is the research question that triggered the analyses conducted. It is stated that: "This study aims to quantify and describe spatial rainfall as a function of temporal and spatial dynamics, rainfall types and seasonal variation". At the preceding paragraph it is written: "Here we apply a broad range of spatio-temporal characteristics in order to develop an automatic classification scheme of different event types and provide a better understanding of actual precipitation processes". These are two different objectives, neither of them mention the word "extreme" which seems to be a main focus. In addition, the Introduction section goes through urban hydrology, design storms, gauge vs. radar rainfall, but not much on topics related to either of the potential objectives mentioned above.

Assuming the main goal is event classification, which also better matches the paper's title, then there are three missing components. The first is validation. I guess the authors could ask some experts to (subjectively) classify the 39 events according to "convective", "frontal" and "others" and test the clustering results against the expert classification. Without validation what we get is cluster analysis with a potential interpretation, not more than that. Second, I would expect a much deeper analysis/discussion of the space-time properties of each type and how is it related to precipitation processes associated with the event type (which was part of the declared objective). Lastly, classifying events is still far from developing an automatic classification scheme, so it is better to remove this part from the goal definition.

**Other comments:**

Event definition is not clear and I have several related questions: 1) the threshold intensity is per pixel or averaged over the area (with size depends on the sampling strategy)? if the former then how does it generalized for the entire area, if the latter then it is a problem because 1 mm/h for 1 km^2 and 38x48km^2 is very different. 2) for what duration the 1 mm/h threshold is applied? I guess that for 5-min, but it is not written clearly. 3) what is the minimal dry duration required for event separation? the authors provided the threshold to separate between dry and wet segment, but can 5-min separate two events? usually a minimal dry duration is set for event separation. 4) why 1 mm/h was set as a threshold? it is described as a "drizzle" threshold, but I think this is not a very low intensity. What percent of 5-min rainfall intensity is above this threshold?

Radar rainfall estimation: I am not clear what Z-R relation was used. It is written that the relation could be described by Z=256R^1.42. Was this the relation used? Another point is that the error reported is on annual basis while the data analyzed are for 15 min, 1h, 24h. So it would be much better to report the error for 24h, using cross-validation procedures (or daily rain gauges that are not participate in the gauge adjustment).

P5,L28: "Extreme events from five independent grid cells are sampled with SS1". In what sense these are independent and how do you know this? surely some events cover more than one pixel out of the five.

P9,L20: "In order to use the knowledge about extreme events from rain gauge data and be able to compare the results obtained to studies using rain gauge data, SS1 is chosen as the sampling strategy for this study". I don't understand why is it important to compare the results to rain gauge? it does not seem a part of any of the potential objectives.

Event selection: in relation to the above point, I am not convinced of the advantage of SS1 sampling strategy. Surely, as shown later in the paper, there are un-sampled extreme events that did not pass over the central sampling pixel with the highest rain intensity. Why not use the entire area?

P10,L1: "It is believed that the most severe extreme events in the case area is sampled for all grid cells, even though the ranking could be different between the grid cells". Why it has to be "believed" and cannot just to be checked? I am not sure this belief is correct. Since only 3 events per year are selected, considering the e-folding correlation distance of 5 km reported later, it can certainly be the case that the largest event in one pixel would be ranked more than 3 in another pixel and would not be selected.

Rain cell properties are not really discussed. It seems like "overkill" to detect and track rain cells without later on relate these properties to storm dynamics.

**Minor comments:**

P2,L2: Another rain gauge strength: direct measurement (much more accurate at the point)

P2,L5: Deriving spatio-temporal properties of the storm: the problem is not with the rain gauge instrument itself but with the rain gauge network that is often too sparse to represent these properties. In principle, a very dense rain gauge network could provide the relevant information on these properties (e.g., the Walnut Gulch gauge network in Arizona).

P.4,L2: 1.4 in percent or as a fraction (i.e., 140%)?

P6,L16: why Dt = 11h?

Rain cell properties are not really discussed

daily (abstract) vs. 24 h

Relevance to urban hydrology is not explored at all, so do not mention in abstract and in introduction

---

## Referee Comment (RC2) · Anonymous Referee #2 · 5 Dec 2020

Review of "Data-driven distinction between convective, frontal and mixed extreme rainfall events in radar data" by Emma Dybro Thomassen et al.

General comments

The manuscript "Data-driven distinction between convective, frontal and mixed extreme rainfall events in radar data" by Emma Dybro Thomassen et al. shows a thorough analysis of rainfall characteristics during extreme events in the Wupper River in Germany. The data and methodology are well exhibited and applied, and are very detailed. This is much needed, as the scientific literature lacks direct treatment of extreme events coming from both high-resolution data and long records. The combination of clustering

analysis with PCA yields interesting results, which can be interpreted as the physical meaning of these statistical approaches, however, some physical understanding and a discussion of its implications are lacking from these analyses (as detailed later). Some general comments are detailed below, and more specific ones are in the other sections of this review.

I recon some minor revisions of this paper are needed before it can be published.

Abstract: To my perspective, the motivation of the study is missing from the abstract. It only appears seemingly as a part of the implications at the end of the abstract. Can you please add a short section in the abstract that describes the motivation or the knowledge gap that led you to perform this thorough analysis? The same holds for the introduction, in which there are some hints about why it is important to characterize extreme events, however to me this part seems lacking.

The discussion about sampling strategies, although interesting, is long, and if at the end of it only SS1 is chosen for the other analyses, I think it would be helpful to describe it more briefly. Similarly, sect. 4.1 can be easily moved to the methods, as the "result" is that SS1 is chosen.

Sect 4.3 can be much improved by showing some representative (or the largest) events. Namely, to my view, it will be better explained by showing one 15-min, one 1-h event and one 24-h event, and describing them shortly (convective / frontal activity, or possibly other kind of event). This could be supplemented by radar QPE maps for accumulated rainfall, or for the maximum precipitation rate of throughout the event. Doing so will also supplement the PC and clustering analyses as you could show where these events are situated with respect to the PC's, and thus in choosing such events you may also consider picking events from the different clusters.

Sect 4.4.1 is interesting, but to my view it lacks some physical inferences. Can you please try to elaborate on the physical meaning of PC1-PC3?

Finally, I am missing some discussion with regards to the motivation of this study, e.g., urban drainage response. Can you please add a small discussion relating the results of this study, especially the clustering part of it, to the motivation?

Specific comments

P2,L15 or L19: consider referring also to Marra and Morin (2015).

P2,L26-30: Please be more specific in your aims. For example, consider adding the study area name to the aims.

P3,L8: "The Bergisches Land is the first major barrier", where do you start counting? Consider adding e.g., "The Bergisches Land is the first major barrier downwind from the North Sea", or something similar. The ending of the sentence is also not clear to me – "Western side" of what? Please write it explicitly.

P4,L20: Please note that some studies typify extremes from spatial measurements based on a large enough amount of pixel passing a threshold (Armon et al., 2020), or based on spatial IDF curves (Rinat et al., 2020).

P5,L28: What do you mean by "independent"? Did you apply some statistical analysis of independence? If not, it is better to say "different", since those cells are probably dependent, at least on an hourly or 24-h timescale.

P9,L3: I am struggling to understand the difference between the fixed number of 39 events that was mentioned earlier, and the number of events cited in Table 2 and in the results section. Please clarify this, and elaborate on the number of events, their definition, and the difference between the 39 and >900 events you mention.

P10,L1: "It is believed" – This could be easily checked. Isn't it?

P10,L9-21: I have the feeling this statement is repeating things that was already mentioned earlier in the paper. It could possibly be better to move these parts together.

Sect 4.2.3: Other comparable results are found also in (Armon et al., 2020) and in

(Marra and Morin, 2018). Fig. 1: Could you please add mean annual precipitation contours to the map? They are described in the text but could be more easily be understood using graphics. To my opinion it would also be beneficial to add the Bergisches Land, the Wuppel River, and the city of Wuppertal to the map.

Fig 2: Consider changing the y-axis to "occurrence frequency".

Technical corrections

P2,L1: "Inferences... two types of data" – These are not the only two types that can be used. Please either add "accurately" (since other products give less accurate data), or "e.g.".

P2,L22: "A improved" should be "An improved".

P2,L27: "A principal component analysis and clustering...", should be "A principal component analysis and a clustering...".

P3,L1, L2 and onward: consider changing "case area" to "study area".

P5,L3: "extend" should be "extent"?

P7,L1: Please change "analysis" to "analyses".

P12,L2: "Ten events is" – please correct to "are".

Table 3: Please add units to the "Similar X" rows. E.g., "[ of events]".

---

## Author Comment (AC1) · 22 Dec 2020

Dear Reviewer, thank you for a thorough review of our article. In the following we have done our best to reply to your comments and suggestions as point to point answers. The review is copied and all our comments start with an asterics to ease reading. ... Major comments: The main difficulty with this study is that it is not clear what is the research question that triggered the analyses conducted. It is stated that: "This study aims to quantify and describe spatial rainfall as a function of temporal and spatial dynamics, rainfall types and seasonal variation". At the preceding paragraph it is written: "Here we apply a broad range of spatio-temporal characteristics in order to develop

an automatic classification scheme of different event types and provide a better understanding of actual precipitation processes". These are two different objectives, neither of them mention the word "extreme" which seems to be a main focus. In addition, the Introduction section goes through urban hydrology, design storms, gauge vs. radar rainfall, but not much on topics related to either of the potential objectives mentioned above.

* We see one of the sentences the reviewer cites as an aim and the other as a means. We will rewrite the introduction with a clearer motivation, aim and methods keeping the above comment in mind as well as comments from the second reviewer.

Assuming the main goal is event classification, which also better matches the paper's title, then there are three missing components. The first is validation. I guess the authors could ask some experts to (subjectively) classify the 39 events according to "convective", "frontal" and "others" and test the clustering results against the expert classification. Without validation what we get is cluster analysis with a potential interpretation, not more than that.

* We have done our best to validate the results by considering the events ourselves but agree with the reviewer that this approach is questionable and that a more independent validation is preferable. A classification by a meteorologist is still subjective, so we suggest instead to make a classification based on the ERA5 reanalysis product that includes variables (e.g. CAPE) that other studies have shown to be important for event classification. We hope that the reviewer agrees such a comparison/validation this will further add to the novelty of our study.

Second, I would expect a much deeper analysis/discussion of the space-time properties of each type and how is it related to precipitation processes associated with the event type (which was part of the declared objective).

* We will elaborate on the event types identified with the cluster method (section 4.4.2, second paragraph) as well as the comparison with ERA5 in order to derive a more

in-depth discussion of the spatio-temporal characteristics. This will also include visualization using the methods shown in Ochoa-Rodriguez et al., 2015, as discussed also with the second reviewer.

Lastly, classifying events is still far from developing an automatic classification scheme, so it is better to remove this part from the goal definition.

* We agree and will rephrase to a "data-driven classification scheme".

Other comments: Event definition is not clear and I have several related questions: 1) the threshold intensity is per pixel or averaged over the area (with size depends on the sampling strategy)? if the former then how does it generalized for the entire area, if the latter then it is a problem because 1 mm/h for 1 km^2 and 38x48km^2 is very different.

* The threshold intensity is per grid cell. When considering more than one grid cell, rainfall occurs when at least one grid cell has an intensity above 1mm/hr. We will clarify this in a revision of the text on P5 L10-16.

2) for what duration the 1 mm/h threshold is applied? I guess that for 5-min, but it is not written clearly.

* Yes; we will clarify this in the revised version of the manuscript.

3) what is the minimal dry duration required for event separation? the authors provided the threshold to separate between dry and wet segment, but can 5-min separate two events? usually a minimal dry duration is set for event separation.

*We use a dry weather period of 24 hours as specified on P5 L7. We understand the comment as a reference to section 3.1 and will move the specification to this section.

4) why 1 mm/h was set as a threshold? it is described as a "drizzle" threshold, but I think this is not a very low intensity. What percent of 5-min rainfall intensity is above this threshold?

* Thank you for this comment. We use a threshold of 1mm/h in the analysis of the

sampling strategies in order to get a high separation into separate events. However, having chosen the sampling strategy SS1 we relax the threshold to a "drizzle" threshold of 1mm/d in the subsequent classification analysis since this is a sufficient threshold to ensure a clear separation into different rainfall types. Table 2 shows number of events with the 1mm/hr threshold for all sampling strategies to be able to compare number of events between strategies while subsequent results use another threshold. We will make this clear in the revised manuscript.

Radar rainfall estimation: I am not clear what Z-R relation was used. It is written that the relation could be described by Z=256R^1.42. Was this the relation used? Another point is that the error reported is on annual basis while the data analyzed are for 15 min, 1h, 24h. So it would be much better to report the error for 24h, using cross-validation procedures (or daily rain gauges that are not participate in the gauge adjustment).

* The Z-R relationship is a function which is variable with the reflectivity, inspired by the one used by the German Weather Service (DWD, 2004). This relationship here, as a difference to the DWD function, uses a true convective formula (Z=256R^1.42) for values above 36 dBZ, and a formula for stratiform rainfall (Z=200R^1.6) below this threshold. Thus, it is considering the type of event, at least in an indirect manner. Thank you for the remark on the reported error: it is indeed described incompletely. The error is counting the number of differences of 5 mm (10 mm) per daily sum over 13 years of data, analysed on independent gauges which were not used for adjustment. We will clarify this in the text.

P5,L28: "Extreme events from five independent grid cells are sampled with SS1". In what sense these are independent and how do you know this? surely some events cover more than one pixel out of the five.

* Thank you for this comment. We will change "independent grid cells" to "different grid cells".

P9,L20: "In order to use the knowledge about extreme events from rain gauge data

and be able to compare the results obtained to studies using rain gauge data, SS1 is chosen as the sampling strategy for this study". I don't understand why is it important to compare the results to rain gauge? it does not seem a part of any of the potential objectives.

* Our study originates out of past research that compares point rainfall estimates with spatial estimates and sampling strategy SS1 is close to this research environment, as reviewer two also points out. We will clarify and justify why this is important as a part of revising the introduction (motivation, aim and methods).

Event selection: in relation to the above point, I am not convinced of the advantage of SS1 sampling strategy. Surely, as shown later in the paper, there are un-sampled extreme events that did not pass over the central sampling pixel with the highest rain intensity. Why not use the entire area?

* We see two established practices for analysis of extreme rainfall; one is based on point rainfall and the other on tracking meteorological events in the atmosphere with little consideration of the catchment below. We have not identified an established practice for sampling of spatial rainfall over a catchment which is the justification of our analysis. Additionally is it common in urban hydrology to use point data which the second reviewer points out should receive more attention in the manuscript. We will make sure to be more accurate in both the justification of studying sampling strategies and the choice of best strategy for the bulk of the analysis.

P10,L1: "It is believed that the most severe extreme events in the case area is sampled for all grid cells, even though the ranking could be different between the grid cells". Why it has to be "believed" and cannot just to be checked? I am not sure this belief is correct. Since only 3 events per year are selected, considering the e-folding correlation distance of 5 km reported later, it can certainly be the case that the largest event in one pixel would be ranked more than 3 in another pixel and would not be selected.

* We agree, and indeed the sampled events will in general vary between grid cells due

to the short e-folding distance. The text refers to the 2-3 most severe extreme events, which are more widespread extremes and hence sampled over a large part of the case area. We will clarify this and check our assumption.

Rain cell properties are not really discussed. It seems like "overkill" to detect and track rain cells without later on relate these properties to storm dynamics.

* The rain cell properties with be used as a part of the missing discussion on physical properties of the different cluster.

Minor comments: P2,L2: Another rain gauge strength: direct measurement (much more accurate at the point)

* Thank you, this will be added.

P2,L5: Deriving spatio-temporal properties of the storm: the problem is not with the rain gauge instrument itself but with the rain gauge network that is often too sparse to represent these properties. In principle, a very dense rain gauge network could provide the relevant information on these properties (e.g., the Walnut Gulch gauge network in Arizona).

* This will be clarified.

P.4,L2: 1.4 in percent or as a fraction (i.e., 140%)?

* We will clarify that we mean 1.4 cases per station per year, equivalent to 0.38% of the days. so extremely rare.

P6,L16: why Dt = 11h?

* We used the same value as reported by Gregersen et al., 2013, that apply the samemethod. We will add a reference to this study here.

Rain cell properties are not really discussed

* We will add this to the discussion on physical properties of the different clusters.

daily (abstract) vs. 24 h

* Thank you, we will make sure this is consistent throughout the manuscript.

Relevance to urban hydrology is not explored at all, so do not mention in abstract and in introduction

* Thank you for the comment. Based on the comments by reviewer 2 we will instead make the links to urban hydrology clearer in the revised manuscript.

Reference Ochoa-Rodriguez S, Wang LP, Gires A, et al (2015) Impact of spatial and temporal resolution of rainfall inputs on urban hydrodynamic modelling outputs: A multi-catchment investigation. J Hydrol 531:389–407. https://doi.org/10.1016/j.jhydrol.2015.05.035

---

## Author Comment (AC2) · 22 Dec 2020

Dear reviewer, we thank you for the very nice and thorough review of our manuscript. In the following we have done our best to reply to your comments and suggestions as point to point answers. The review is copied and all our comments start with an asterics to ease reading.

...

Abstract: To my perspective, the motivation of the study is missing from the abstract. It only appears seemingly as a part of the implications at the end of the abstract. Can

you please add a short section in the abstract that describes the motivation or the knowledge gap that led you to perform this thorough analysis? The same holds for the introduction, in which there are some hints about why it is important to characterize extreme events, however to me this part seems lacking.

\* Thank you, we will rewrite the abstract and introduction with a clearer motivation and aim of the study.

The discussion about sampling strategies, although interesting, is long, and if at the end of it only SS1 is chosen for the other analyses, I think it would be helpful to describe it more briefly. Similarly, sect. 4.1 can be easily moved to the methods, as the "result" is that SS1 is chosen.

\* We will try to be clearer in why this discussion is necessary yet still condense the relevant information. We suggest to put most information into a table for a quicker and shorter overview. We hope these changes makes it clearer why we would like to keep the 4.1 result section (also see later comment about discussion on urban drainage).

Sect 4.3 can be much improved by showing some representative (or the largest) events. Namely, to my view, it will be better explained by showing one 15-min, one 1-h event and one 24-h event, and describing them shortly (convective / frontal activity, or possibly other kind of event). This could be supplemented by radar QPE maps for accumulated rainfall, or for the maximum precipitation rate of throughout the event. Doing so will also supplement the PC and clustering analyses as you could show where these events are situated with respect to the PC's, and thus in choosing such events you may also consider picking events from the different clusters.

\* Thank you for the suggestion. We will add a figure with a typical event from each of the four clusters. We anticipate to use a layout corresponding to figure 2 in Ochoa-Rodriguez et al., 2015.

Sect 4.4.1 is interesting, but to my view it lacks some physical inferences. Can you

please try to elaborate on the physical meaning of PC1-PC3?

* We will elaborate on the physical understanding of PC1-PC3 and also try to relate it to a new section on validation of the classification against ERA5.

Finally, I am missing some discussion with regards to the motivation of this study, e.g., urban drainage response. Can you please add a small discussion relating the results of this study, especially the clustering part of it, to the motivation?

* Yes, we will make sure the urban drainage aspect is proper discussed together with the results of the sampling methods and the clustering results.

Specific comments P2,L15 or L19: consider referring also to Marra and Morin (2015).

* Thank you for suggesting this reference, we will refer to it in L19.

P2,L26-30: Please be more specific in your aims. For example, consider adding the study area name to the aims.

* We will make sure the aim is clearer when we revise the abstract and introduction. As the aim of the manuscript is not dependent on the study area, we will not include the name here.

P3,L8: "The Bergisches Land is the first major barrier", where do you start counting? Consider adding e.g., "The Bergisches Land is the first major barrier downwind from the North Sea", or something similar. The ending of the sentence is also not clear to me – "Western side" of what? Please write it explicitly.

* Thank you for your suggestion, we will add that and make sure to clarify the ending of the sentence.

P4,L20: Please note that some studies typify extremes from spatial measurements based on a large enough amount of pixel passing a threshold (Armon et al., 2020), or based on spatialIDF curves (Rinat et al., 2020).

\* Thank you for the suggested articles. We will consider these articles as examples of spatial sampling strategies.

P5,L28: What do you mean by "independent"? Did you apply some statistical analysis of independence? If not, it is better to say "different", since those cells are probably dependent, at least on an hourly or 24-h timescale.

\* Thank you, we will change the sentence to "Extreme events from five different grid cells".

P9,L3: I am struggling to understand the difference between the fixed number of 39 events that was mentioned earlier, and the number of events cited in Table 2 and in the results section. Please clarify this, and elaborate on the number of events, their definition, and the difference between the 39 and >900 events you mention.

\* The +900 events is the pool of events (after applying a drizzle threshold and 24 dry weather separation) from which the 39 extreme events are sampled from. We will make sure this is stated clearly.

P10,L1: "It is believed" – This could be easily checked. Isn't it?

\* Thank you, this will be checked.

P10,L9-21: I have the feeling this statement is repeating things that was already mentioned earlier in the paper. It could possibly be better to move these parts together.

\* We see your point, the discussion on sampling strategies are started in section 4.1 and continued in 4.2.1, resulting in an overlap. We will shorten the discussion in section 4.1 and refer to the discussion in 4.2.1, to avoid repeating statements

Sect 4.2.3: Other comparable results are found also in (Armon et al., 2020) and in (Marra and Morin, 2018).

\* Thank you for pointing us towards these articles. We will add them to the discussion,

[Figure]

Fig. 1: Could you please add mean annual precipitation contours to the map? They are described in the text but could be more easily be understood using graphics. To my opinion it would also be beneficial to add the Bergisches Land, the Wuppel River, and the city of Wuppertal to the map.

* We will add mean annual precipitation contours to the map and try to make the background image clearer, so the reader is able to see the Wupper River and city of Wuppertal on the map.

Fig 2: Consider changing the y-axis to "occurrence frequency".

* We guess this applies to Fig 3, instead of Fig 2. We will change the y-axis to make it more understandable.

Technical corrections

* Thank you for the 8 technical corrections, we will make sure to implement those.

References

Armon M, Marra F, Enzel Y, et al (2020) Radar-based characterisation of heavy precipitation in the eastern Mediterranean and its representation in a convection-permitting model. Hydrol Earth Syst Sci 24:1227–1249. https://doi.org/10.5194/hess-24-1227-2020

Deutscher Wetterdienst (2004): Projekt RADOLAN – Routineverfahren zur Online-Aneichung der Radarniederschlagsdaten mit Hilfe von automatischen Bodenniederschlagsstationen (Ombrometer); Abschlussbericht, 111 Seiten

Ochoa-Rodriguez S, Wang LP, Gires A, et al. (2015) Impact of spatial and temporal resolution of rainfall inputs on urban hydrodynamic modelling outputs: A multi-catchment investigation. J Hydrol 531:389–407. https://doi.org/10.1016/j.jhydrol.2015.05.035

Marra F, Morin E (2015) Use of radar QPE for the derivation of Intensity-Duration-Frequency curves in a range of climatic regimes. J Hydrol 531:427–440.

[Figure]

https://doi.org/10.1016/j.jhydrol.2015.08.064

Marra F, Morin E (2018) Autocorrelation structure of convective rainfall in semiarid-arid climate derived from high-resolution X-Band radar estimates. Atmos Res 200:126–138. https://doi.org/10.1016/j.atmosres.2017.09.020

Rinat, Y., Marra, F., Armon, M., Metzger, A., Levi, Y., Khain, P., Vadislavsky, E., Rosensaft, M., and Morin, E.: Hydrometeorological analysis and forecasting of a 3-day flash-flood-triggering desert rainstorm, Nat. Hazards Earth Syst. Sci. Discuss., https://doi.org/10.5194/nhess-2020-189, in review, 2020.
* * *

---

## Referee Comment (RC3) · Anonymous Referee #3 · 15 Jan 2021

Review of the manuscript titled "Data-driven distinction between convective, frontal and mixed extreme rainfall events in radar data" submitted to HESS

**Paper overview / general comments:**

This manuscript developed a method to categorise extreme storm events using high-resolution radar images in Germany. The selected extreme storm events were characterised with a total of 17 storm features; and the PCA (Principle Component Analysis) technique was used to reduce the dimensions to 5 – 9. Finally, the k-mean clustering algorithm was used to classify storm types according to the PCA outcome.

The overall organisation of the manuscript is not great. In particular, the link is weak between the proposed sampling strategies for characterising spatial dependence of extreme events and the storm type classification. The section 3.2 does not make much sense to me, and the proposed strategies are not convincing. Especially, a well-established geostatistically-based method to quantify the spatial dependence of storm events has been developed in Ochoa-Rodriguez et al. (2015) (see Section 3.2.1). I would encourage the authors to have a detailed look at the method mentioned above.

In addition, in section 4.2, the authors looked into both spatial variation and spatial correlation. The former was investigated according to the proposed sampling strategies, whilst the latter was performed using all radar pixels. In my opinion, these two are very similar characteristics. I am not convinced why they have to be done over different domains.

Moreover, a lot of efforts were made to explain the sampling strategies and spatial structure of storms, but it seems no 'spatial' features were included in the PCA analysis. This is really strange to me. I wonder if the authors can explain the reason excluding spatial features in the PCA analysis and consequently storm classification.

Finally, it is pity that the authors did not cross compare the results obtained from the proposed classification method and those from some widely-used methods (such as Steiner et al., 1995; Biggerstaff and Listemaa, 2000). I believe this would provide more insights about the quality of the proposed classification method.

From my point of view, the authors have to address the above issues before it can be published. Given that these issues may be difficult / time consuming to address, I suggest either very major corrections or that the manuscript be rejected in its current form.

Minor comments:

- Page 3, Lines 29: were the radar reflectivity data of all selected extreme events converted into rainfall rates using the Z-R relationship specified here? What is the Z-R relationship used for frontal (or stratiform) events?
- Page 4, Line 10: Could you please explain why 39 extreme events?
- Page 4, Line 24: What do you mean by 'grid cells registering each event'?
- Page 7, Line 7: In terms of tracking algorithm, I would recommend to include Muñoz et al., 2018.

References

Biggerstaff, M. I. and Listemaa, S. A., 2000: An improved scheme for convective/stratiform echo classification using radar reflectivity, Am. Meteorol. Soc. 39, 2129-2131.

Muñoz, C., et al., 2018: Enhanced object-based tracking algorithm for convective rain storms and cells, Atmos. Res., 201, 144-158.

Ochoa-Rodriguez, S., et al., 2015: Impact of spatial and temporal resolution of rainfall inputs on urban hydrodynamic modelling outputs: A multi-catchment investigation, J. Hydrol., 531, 389-407.

Steiner, M., et al., 1995: Climatological characterization of three-dimensional storm structure from operational radar and rain gauge data, J. Appl. Meteorol. 34: 1978-2007.